# Impact of Telemedicine Lecture on Online Medical Interview Performance

**Michael W. Myers** [1,*]**, Kris Siriratsivawong** [2]**, Yoshiko Kudo** [1]**, Yuka Hiraizumi** [1] **and Miyuki Hashimoto** [1]

1    International Exchange Center, Showa University, 1-5-8 Hatanodai Shinagawa-ku, Tokyo 142-8555, Japan
2    Department of Medical Education, School of Medicine, Showa University, 1-5-8 Hatanodai Shinagawa-ku, Tokyo 142-8555, Japan
*    Correspondence: myersmw@dent.showa-u.ac.jp; Tel.: +81-(3)-3784-8266

**Abstract:** In 2019, Showa University implemented a compulsory clinical English course for all 4th-year medical students that included a medical interview with an English-speaking standardized patient (ESSP), but since 2020 the interviews have been conducted online due to the novel coronavirus pandemic. These students reported difficulties with eye contact and reading body language/nonverbal cues of the ESSP. In this project report, we describe a telemedicine lecture that we included in the 2021 course and compare students' reported difficulties during the online medical interview for two years. The 2021 cohort reported significantly less difficulties with eye contact than the 2020 cohort, and a similar trend was found for reading body language/nonverbal cues and creating rapport with the ESSP. While possible third variables, such as 2021 cohort's greater comfort in using teleconference platforms, may limit the interpretation of these results, we conclude that Japanese medical students can clearly benefit from the inclusion of telemedicine education into the medical curriculum as online healthcare services become the "new normal" in Japan.

**Keywords:** telemedicine; medical interview skills; medical English education; curriculum development

## 1. Introduction

As a result of students' growing interest in pursuing medical training outside of Japan [1], and a desire to prepare students for medicine that is becoming increasingly global [2–4], Japanese medical schools have begun to further expand English education into their curriculum. Currently, the Japanese medical education system is a six-year long program that students enter after completing high school. Medical students typically take general education classes (e.g., biology, chemistry, English, etc.) during their 1st year and basic/general medical science classes (e.g., anatomy, immunology, pathology, etc.) during their 2nd to 3rd years [5]. After passing OSCE, which assesses clinical skills and bedside manner, along with a computer-based test of knowledge during their 4th year, medical students can then engage in clinical training in the hospital for the rest of their 4th year and final years (5th and 6th year) [6]. Until recently, Showa University (SU) School of Medicine, a medical school located in Tokyo with a total enrollment of 720 students in 2020, only included medical English education up to the 3rd year, and the focus was primarily on basic reading and speaking skills.

However, a new curriculum was introduced at SU for the freshmen starting with the 2020 academic year, which also provided our medical school the opportunity to revise the English education throughout students' six years in the medical program so that it can successfully prepare students for providing medicine that is increasingly global and standardized. Specifically, in 2019 Showa University School of Medicine implemented a new compulsory clinical English course for all 4th-year medical students, called Medical English for Clinical Purposes A. The goal of this course is for students to acquire advanced oral and written English communication skills in the clinical setting so that they can be actively engaged in healthcare globally. Thus, a main component of this course is that

all students (approximately 120 each year) conduct an 8 min medical interview with an English-speaking standardized patient (ESSP) as part of their final assessment.

Unfortunately, due to the novel coronavirus pandemic, the course was transitioned to an online format in 2020. Lectures became pre-recorded videos that were uploaded to a "virtual classroom" through Google Classroom where students could view them on demand. Furthermore, the medical interview with ESSP assessment was held online using Zoom. This change to an online format has created both benefits and problems for our students. We have previously reported that most students from the 2020 course preferred this online format over traditional in-person classes, primarily because they could view the teaching materials at any time and watch lecture videos more than once. However, conducting the medical interview online was rated less favorably, with students reporting challenges with maintaining appropriate eye contact and difficulty in reading body language/nonverbal cues from the patients [7]. Other studies on telemedicine have noted similar difficulties when medical students first use this new format to communicate with patients [8,9].

Given these noted challenges, for the 2021 course we added a lecture on tips for telemedicine to help prepare students for the final assessment as well as to provide a brief introduction of this topic to their medical education. Telemedicine is an increasingly important component of healthcare delivery, with an expected market size of $176 billion by 2026 [10]. It is imperative that medical schools teach students the vital skills to deliver health care services using digital communication technology. Several U.S. medical schools have already started to integrate telemedicine education into their curriculum [11,12]. In contrast, not as much progress has been made in Japan, and in fact, we know of no study regarding the experience of medical schools in teaching telemedicine.

In this project report, we summarize the telemedicine lecture that we provided to the 2021 students, and then compare 2020 and 2021 students' reported difficulties with the online medical interview to assess if this lecture helped prepare students.

## 2. Detailed Case Description

### 2.1. Summary of Student Cohorts

Participants were 4th-year medical students at Showa University who took a compulsory course on clinical English, titled "Medical English for Clinical Purposes A", in 2020 or 2021 and completed a post-course online questionnaire about its conversion to an online format. Of the 123 students in the 2020 course, 117 (95.1%) completed this post-course questionnaire. There were 75 men and 42 women (mean age = 22.9, range 21–30 years). Of the 124 students in the 2021 course, 121 (97.6%) completed this post-course questionnaire. There were 89 men and 32 women (mean age = 22.9, range 21–36 years). Fisher exact tests indicated no significant differences in the completion percentage nor gender distribution between the 2020 and 2021 cohorts. Students originally completed the questionnaire as part of a course requirement, but afterwards we received approval from Showa University Research Ethics Review Board to publish this data for research purposes, pending consent from the students (approval code 3326). All students were notified of this change and were given the opportunity to opt out by requesting that their data not be used for this research.

### 2.2. Summary of Telemedicine Lecture

The telemedicine lecture was delivered in a PowerPoint format that contained about 12 slides, and as a pre-recorded audiovisual lecture, which was about five minutes in length. The content was loosely based on a variety of available online resources, including practice tips for presenting at home, as provided by the Radiological Society of North America [13]. The following topics and details were included in the lecture content:

- Eye contact: Maintaining eye contact at or near the camera would give the semblance of looking directly at the patient.
- Camera angle and position: Keeping the camera and screen at eye level would promote a natural position for viewing the patient. Cameras that are too low in position will

lead to an unnatural angle from which the patient views their doctor's face, with the ceiling being the unintended background.

- Background: A background that is uncluttered and neat-looking would provide a comfortable environment for the patient.
- Framing: Ensuring that the patient can see the head and upper body would allow them to understand the necessary body language, allowing communication using hands.
- Lighting: Ensure the light is projected onto the face, rather than from above or behind.
- Sound: It is important to test the audio in advance and to speak clearly.
- Microphone: A good quality microphone would promote improved audio and ensure clear verbal communication.
- Appropriate attire: A professional appearance should be maintained, even in the virtual environment.

An online questionnaire was administered through the course's Google Classroom at the end of the course. Students evaluated several aspects of the course, including their perception of the class as a whole, the online delivery of the course, and the online delivery of the medical interview with ESSP.

### 2.3. Comparison of Cohort Responses

Independent-samples *t*-tests and Pearson chi-square tests were used to identify potential differences in the two cohorts. These data are found in Table 1. Regarding students' perception of the course, both *t*-tests and chi-square tests indicated that the 2021 cohort reported significantly higher confidence in providing medical care to the English-speaking patient at the end of the course. The *t*-test also showed that, compared to the 2020 cohort, the 2021 cohort rated the course as significantly easier, although this comparison did not reach significance in the chi-square test. Regarding the online medical interview, students' ratings of the effectiveness of the medical interview with ESSPs was similar for the two cohorts, with neither *t*-test nor chi-square test being significant; however, both tests indicated that the 2021 cohort preferred the online format of the medical interview to the in-person interview significantly more than the 2020 cohort.

**Table 1.** Comparison of responses about course and online medical interview between 2020 and 2021 cohorts.

| Item | Year | % for Each Likert-Scale Rating | | | | | Mean (SD) | $\chi^2$ Score | t Score |
|---|---|---|---|---|---|---|---|---|---|
| | | **1** | **2** | **3** | **4** | **5** | | | |
| The level of difficulty was: † | 2020 | 12.2 | 40.0 | 45.2 | 1.7 | 0.9 | 2.39 (0.76) | 9.0 | 2.8 ** |
| | 2021 | 8.3 | 26.7 | 56.7 | 5.8 | 2.5 | 2.68 (0.81) | | |
| For the medical care of English-speaking patients, my confidence level NOW is: | 2020 | 27.4 | 35.9 | 34.2 | 1.7 | 0.9 | 2.13 (0.87) | 20.9 *** | 4.4 *** |
| | 2021 | 11.7 | 33.3 | 37.5 | 10.8 | 6.7 | 2.68 (1.04) | | |
| The use of Standardized Patients for this session was effective for improving my clinical history taking skills. | 2020 | 1.7 | 6.8 | 12.8 | 61.5 | 17.1 | 3.85 (0.84) | 5.2 | 1.8 |
| | 2021 | 0.0 | 4.2 | 13.3 | 56.7 | 25.8 | 4.04 (0.75) | | |
| Compared to a traditional in-person clinical interview, I prefer the online method of clinical interview via Zoom: | 2020 | 6.8 | 19.7 | 35.0 | 29.1 | 9.4 | 3.15 (1.06) | 21.8 *** | 4.7 *** |
| | 2021 | 1.7 | 5.0 | 31.4 | 42.1 | 19.8 | 3.74 (0.89) | | |

** $p < 0.01$; *** $p < 0.001$. † The rating scale for this item was from "too hard"(1) to "too easy"(5).

### 2.4. Downsides of Online Medical Interview

Our primary interest was in students' reports of downsides of the online medical interview and whether the lecture on telemedicine led to a reduction in difficulties in maintaining appropriate eye contact and reading body language/nonverbal cues from the patients for the 2021 cohort. Table 2 shows the proportion of students who selected each of the nine downsides of the online medical interview for the 2020 and 2021 cohorts. To examine if these proportions were significantly different between the two cohorts, we ran a chi-square test for each item. These tests revealed that the proportion of students who reported difficulty with eye contact during the online medical interview was significantly

lower in the 2021 cohort compared to the 2020 cohort, $\chi^2(1) = 6.26$, $p < 0.05$. As shown in Table 2, difficulty reading body language and difficulty building rapport with the ESSP also showed a similar trend, with the proportion of students in the 2021 cohort reporting these difficulties at a much lower percentage (7% and 6% lower, respectively) than the 2020 cohort; however, neither of these differences were significant. For the other difficulties (feeling nervous because it was a first-time experience, schedule being complicated, difficulty preparing appropriate space at home, etc.), none of the chi-square tests were significant and the difference in proportions between the two cohorts was smaller (range 0.1–3.3%).

**Table 2.** Proportion of students who selected each downside of the online medical interview for 2020 and 2021 cohorts.

| Downside | 2020 Cohort % ($n = 117$) | 2021 Cohort % ($n = 121$) | $\chi^2$ Score | *p*-Value |
|---|---|---|---|---|
| Schedule was complicated. | 13.7 | 14.0 | 0.007 | 0.93 |
| First-time experience, so I was nervous/confused. | 54.7 | 53.7 | 0.023 | 0.88 |
| Did not know how to do eye contact. | 63.2 | 47.1 | 6.26 * | 0.012 * |
| Difficulty preparing appropriate space at home for interview. | 25.6 | 22.3 | 0.361 | 0.55 |
| Difficulty reading body language and nonverbal cues from SP. | 27.4 | 19.8 | 1.87 | 0.17 |
| Difficulty creating rapport with SP. | 30.8 | 21.5 | 2.66 | 0.10 |
| Embarrassed/Uncomfortable seeing my face during the interview. | 12.8 | 11.6 | 0.087 | 0.77 |
| None of the above. | 3.4 | 6.6 | 1.27 | 0.26 |
| Other. | 3.4 | 3.3 | 0.002 | 0.96 |

* $p < 0.05$.

## 2.5. Technical Difficulties of Online Clinical Interview

Finally, we examined if there were any differences in the proportions of technical problems encountered during the online medical interview. As Table 3 shows, only the "slow transmission/time lag" problem showed a significant difference between the cohorts. Specifically, the 2020 cohort reported this difficulty significantly more frequently than the 2021 cohort.

**Table 3.** Proportion of students who selected each of the technical problems during the online medical interview for 2020 and 2021 cohorts.

| Technical Difficulty | 2020 Cohort % ($n = 117$) | 2021 Cohort % ($n = 121$) | $\chi^2$ Score | *p*-Value |
|---|---|---|---|---|
| Unable to enter the Zoom meeting | 2.6 | 0.8 | 1.09 | 0.30 |
| Poor video quality | 6.8 | 8.3 | 0.173 | 0.68 |
| Poor audio quality | 20.5 | 14.9 | 1.30 | 0.25 |
| Slow transmission/Time lag | 29.9 | 13.2 | 9.84 | 0.002 * |
| Echo from room | 5.1 | 3.3 | 0.491 | 0.48 |
| I experienced no technical problems | 55.6 | 66.9 | 3.25 | 0.07 |
| Other | 0.0 | 0.4 | 0.971 | 0.32 |

* $p < 0.05$.

## 3. Discussion

During the 2020 course, our students experienced difficulties conducting an online medical interview, notably making eye contact and reading body language/nonverbal cues from the patients, which we attributed to their inexperience with telemedicine. Therefore, in 2021 we added a new lecture that introduced the basic concept of telemedicine and provided pointers for improving communication skills. The purpose of the telemedicine lecture was to provide a guide for students to ultimately develop an improved rapport with patients in the teleconference environment. The topics included how to make eye contact, including angling the camera and positioning the camera at a height that ensures proper framing of the medical provider's face. Lighting is also important, as this ensures that

facial expressions are easily visible. Testing the sound and microphone in advance ensures that the interview can proceed smoothly. Appropriate attire and a good background that is uncluttered will create a good environment for the patient interview.

In this project report, we found that a significantly fewer proportion of students in the 2021 cohort reported difficulties with eye contact during the online medical interview than the 2020 cohort, and the proportion of students reporting difficulties with reading the body language and forming a rapport with the ESSP was also markedly lower in 2020 compared to 2021. We interpret these results as promising, although indirect, evidence that the introduction of a telemedicine lecture helped our students prepare for this form of online healthcare. From the 2022 academic year, we have included a question in the post-course questionnaire that more directly assesses a student's perception of the usefulness of the course in helping them to practice telemedicine in the future.

However, a major limitation of this report is that there may be some unaccounted-for differences between the two cohorts that explain our findings, rather than the introduction of our telemedicine lecture. For example, while the response rate and gender distribution were similar between the two cohorts, we did find that the 2021 cohort preferred the online format of the medical interview significantly more than the 2020 cohort. With the novel coronavirus pandemic soon approaching its third year, the 2021 cohort may have had more experiences with the online course/meeting format than the 2020 cohort. Consequently, the 2021 cohort generally may have been more comfortable with online interactions, including the use of Zoom and other teleconference platforms, and these students may have previously learned how to avoid some online communication problems on their own. The fact that the 2021 cohort reported less technical problems with slow transmission and time lag may suggest a better proficiency with online communication, although, interestingly, the other technical difficulties were not different between the cohorts. Additionally, the 2021 cohort reported that the course was less difficult and that they had greater confidence treating English-speaking patients at the end of the course, which may suggest that they had a higher English communication ability than the 2020 cohort. Students' better English communication abilities may have helped them overcome initial challenges of the online interview format that would normally be problematic for other students of lower English ability. Consequently, the cut-off point at which eye contact or nonverbal issues became difficulties of patient communication may have been set higher in the 2021 cohort than the 2020 cohort.

## 4. Conclusions

We think formal education on telemedicine should be an integral part of a medical student's undergraduate training in Japan and elsewhere, and we hope to expand it from this medical English course and into other areas of the curriculum. The novel coronavirus pandemic has highlighted the importance of telemedicine and accelerated its use in the field. Due to the pandemic, Japan's Ministry of Health, Labor and Welfare relaxed restrictions on the use of telemedicine, making it more readily available [14]. Consequently, the number of medical facilities offering telemedicine services in Japan jumped 60%, from 10,624 (9.6%) in April 2020 to 16,814 (15.1%) in April 2021 [15,16]. We believe that a "new normal" in the Japanese medical field will be an increased reliance on telemedicine to provide high-quality care to patients.

**Author Contributions:** Conceptualization, M.W.M., K.S., Y.K., Y.H. and M.H.; methodology, M.W.M., K.S., Y.H. and M.H.; formal analysis, M.W.M.; investigation, M.W.M., K.S., Y.K., Y.H. and M.H.; data curation, M.W.M.; writing—original draft preparation, M.W.M., K.S., Y.K., Y.H. and M.H.; writing—review and editing, M.W.M., K.S., Y.K., Y.H. and M.H.; visualization, M.W.M.; supervision, M.W.M. and M.H.; project administration, M.W.M. and M.H.; All authors have read and agreed to the published version of the manuscript.

**Funding:** This research received no external funding.

**Institutional Review Board Statement:** The study was conducted in accordance with the Declaration of Helsinki, and approved by the Institutional Review Board of Showa University (protocol code 3326, 20 June 2021).

**Informed Consent Statement:** Informed consent was obtained from all subjects involved in this study.

**Data Availability Statement:** The data presented in this study are available on request from the corresponding author. The data are not publicly available due to privacy issues.

**Conflicts of Interest:** The authors declare no conflict of interest.

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
