# Peer review of "Impact of Telemedicine Lecture on Online Medical Interview Performance"

_ime, doi:10.3390/ime1010005_

Round 1

Reviewer 1 Report

The authors clearly report their pioneering experience of a telemedicine module introduced during the COVID-19 pandemic.

I have no major concerns on the manuscript and one suggestion for further improvement:

- the authors should consider adding a short paragraph highlighting how their medical degree program is structured and the total number of attending students. This would help international readers to put the results into an international perspective.

Author Response

Thank you very much for the positive feedback to our manuscript. Attached is our revised manuscript. Below, we will briefly describe our changes.

1. Per your request, we included additional information about the Japanese medical degree program and the total number of attending students in our medical school. These edits are located in the first paragraph of the Introduction.

2. Although we had intended to submit this manuscript as an "educational case report", we discovered that its structure didn't follow the appropriate format. Thus, we have reorganized/edited the information in our revised manuscript and renamed our sections so that its structure now correctly follows the educational case report format (Introduction, Detailed Case Report, Discussion, Conclusion). We apologize for any confusion during the reviewing process!

Reviewer 2 Report

The authors summarize the effects of a telemedicine lecture provided to students in 2021 and, to assess whether this lecture helped prepare students better, compared the online medical interview difficulties reported by students in 2020 vs 2021. The manuscript is not badly structured, it certainly lacks in methodological quality, so I recommend a major revision of the manuscript.

Major Comments

1. The main limitation of the study is the lack of a control group. Many variables may have contributed to providing the result of this study. Above all, the massive use of online methods that we all experienced between 2020 and 2021 has changed not only our perceptions but also our skills. Therefore, without a control group, the results discussed in the present work cannot be generalized. Please add a control group to confirm these results.

2. The authors wanted to measure the EFFECTIVENESS of the telemedicine lecture. Effectiveness is a complex construct, therefore it is simplistic to measure it only with a question without considering the indicators of the construct.

3. Benefit of online format results are not shown. Please add these results.

4. Questionnaire, if possible, should be fully provided (in the article, or as appendices or as an online supplement).

5. Increase the quality of the tables for example, mean + -SD and n (%).

Minor Comments

Line 126: “However” it can be written in a lowercase letter.

Line 183: …..in 2021 compared to 2021. There is a typo.

Author Response

Thank you for your thorough review of our manuscript. As an original research article, we completely agree with your assessment that there are several major limitation in this study, notably that there is no control group and that our constructs were measured rather simplistically.

However, our actual purpose was to submit this manuscript as an "educational case report" instead of original research. Unfortunately, we realized that the format of our original manuscript didn't match the appropriate structure of an educational case report. Thus, we believe there was confusion for the reviewers. 

In the attached file, we have reorganized/edited the information and renamed the section headings in our revised manuscript so that it now correctly follows the structure of an educational case report (Introduction, Detailed Case Description, Discussion, Conclusion). As an educational case report, we hope the reviewer will reconsider the major limitations he/she previously wrote and decide that they're not as relevant for an educational case report. However, we have corrected the 2 minor comments included in the review. 

Round 2

Reviewer 2 Report

I thank the authors for the revisions of the text, however I believe that the critical issues found in the previous version of the manuscript have not been resolved in new version. In my opinion the results of the study are not adequately supported by the data, therefore I would not accept this article for publication.